# Application of Sliding Rehabilitation Machine in Patients with Severe Cognitive Dysfunction after Stroke

**Ae Ryoung Kim [1,2] and Yang-Soo Lee [1,2,\*]**

[1]  Department of Rehabilitation Medicine, School of Medicine, Kyungpook National University, Daegu 41944, Korea; ryoung20@hanmail.net

[2]  Department of Rehabilitation Medicine, Kyungpook National University Hospital, Daegu 41944, Korea

\*  Correspondence: leeyangsoo@knu.ac.kr; Tel.: +82-53-200-5311; Fax: +82-53-200-5311



**Featured Application: This study illustrates that the use of a sliding rehabilitation machine (SRM) for early intensive muscle strengthening is applicable to patients who have had a stroke with severe cognitive dysfunction and demonstrates that the SRM is safe when used as part of an inpatient rehabilitation program.**

**Abstract:** A sliding rehabilitation machine (SRM) allows closed kinetic chain exercises of the hip, knee, and ankle. This study aimed to explore the feasibility of SRM training when included in an intensive rehabilitation program for post-stroke patients with severe cognitive dysfunction. The study design is a retrospective analysis. Patients who were admitted for inpatient rehabilitation after stroke with subsequent severe cognitive dysfunction were enrolled. Training with the SRM was conducted twice a day from Monday to Friday during hospitalization for three to four weeks. The number of sessions and the occurrence of side effects were documented daily. The SRM's inclination angle, Berg Balance Scale (BBS), manual muscle test (MMT), and Korean version of the Modified Barthel Index (K-MBI) were documented upon admission and discharge. In 30 patients, 1736 sessions were performed from a total of 1754 scheduled sessions of SRM training. The performance rate was 98.9%, and there were no serious side effects. Transient side effects such as dizziness, nausea, and knee pain were observed in a few cases. At discharge, patients showed improvement in the SRM inclination angle, BBS, MMT, and K-MBI. This study shows that the use of the SRM for intensive muscle strengthening is readily applicable to patients who have had a stroke with severe cognitive dysfunction.

**Keywords:** post-stroke patient; cognitive dysfunction; strengthening; rehabilitation

---

## 1. Introduction

Stroke is a common cause of severe disability worldwide and a global health care problem [1,2]. Most patients who have had a stroke have varying degrees of neurological deficit ranging from mild to severe [3]. Representative neurological impairments include motor weakness, swallowing difficulty, as well as sensory and cognitive impairment [4]. Proper rehabilitation may reduce disability and help to restore neurological impairment [2,5]. A primary goal for patients after a stroke is to restore deficits in gait [2,6]. Lower extremity muscle strength and cognitive function are important determinants of the recovery of gait ability [7,8]. In the case of severe cognitive impairment, cooperation with the therapist is limited, and in many cases, the use of an apparatus is restricted [9,10]. In patients who have had a stroke with severely impaired cognitive function, the difficulty in participating in active rehabilitation due to the decrease in cooperation is the main reason of the poor outcome [9,11].

Applying an appropriate treatment to these patients and simultaneously achieving cognitive and motor function recovery represent a big challenge.

The most important factor in the recovery of motor performance of patients who have had a stroke is the active involvement of the patients themselves through task-oriented or goal-oriented training [2,12]. Recently, various treatment methods and robots have been developed and applied in practice [13], but their application is very limited for patients suffering from neurological damage and severe cognitive decline. Moreover, leg strengthening is the most basic rehabilitation treatment for patients to stand on their own and try to recover their ability to walk [14]. Weight-bearing training is also an important technique for gait training [15]. Based on this, recovery of the gait function is speculated possible in the future. However, in most patients with severe paralysis and cognitive impairment, no treatment can be applied other than to induce patient motion of the hands on a Bobath table.

A sliding rehabilitation machine (SRM) (Conble LS, Kwangwon Meditec, Busan, South Korea) [16,17], approved by Korea Food & Drug Administration (KFDA), has the advantage of providing both weight-bearing exercises and forced repetitive flexion and extension of the hemiplegic lower extremities. SRM training can be applied to patients according to their muscle strength and gravity response by adjusting the inclination of the machine. Because SRM training involves the repetition of a simple knee flexion and extension motion, we hypothesized that it could be further applied to rehabilitation programs with patients who have had a stroke with severe cognitive impairment. To our knowledge, no study has applied SRM training to post-stroke patients with such an impairment. Therefore, this study aimed to explore the safety and feasibility of the SRM system in intensive rehabilitation programs for post-stroke patients with severe cognitive impairment.

## 2. Materials and Methods

This retrospective study was conducted at the department of rehabilitation medicine of a university hospital. Participants were patients who were admitted after stroke between June 2014 and January 2018. The patients received an integrated, individualized, and team-based inpatient rehabilitation program. The inclusion criteria were (1) diagnosis of cerebral infarction and (2) presence of severe cognitive impairment, that is, a Korean version of the Mini-Mental State Examination (K-MMSE) value of ≤9 [18,19]. The exclusion criteria were (1) presence of severe visual impairment and (2) other neurologic or musculoskeletal impairment with inability to perform using the SRM. Therefore, the patients who were enrolled this study had severe cognitive impairment.

### 2.1. Training Program

The SRM consists of a rail system, a carriage, and a footplate to support the patient. By moving the stop bar—which manipulates the angle of carriage—up or down, the physical therapist can change the patient's angle of knee flexion and extension. The leg support is connected to the carriage that can support the non-hemiplegic lower extremity (Figure 1). The patient lies on the parallel carriage to exercise, that is, repetitive knee flexion and extension. Patients can perform SRM training using bilateral lower extremities or a unilateral lower extremity (Figure 2).

The goniometer on the side of the instrument indicated the angle of the sliding board on which the patient was trained and ranged from 0 to 90° depending on the patient's lower extremity strength. Velcro straps were used to fix the patient's body and ankles for safety. The inclination angle was set to change daily to the appropriate inclination at which patients could actively exercise with lower extremities flexion and extension. Therefore, it was possible to gather quantitative data, such as appropriate inclination and maximum number of repetitions of the training at a given inclination.

Training with the SRM was performed during two sessions daily, from Monday to Friday during the admission period. The patients were encouraged to repeat the exercise 100 times. In all patients, the training was performed in addition to conventional physical therapy. The inclination angle of the SRM was individualized. If the patient could repeat the exercise 100 times at apredetermined angle (i.e., the patient has enough strength to exercise the angle), the physical therapist could then increase

the carriage's angle by 1° at the next session. At the beginning of the initial training, knee flexion and extension were manually guided, and then the patient was allowed to perform independently. All training sessions were conducted by an experienced physical therapist.

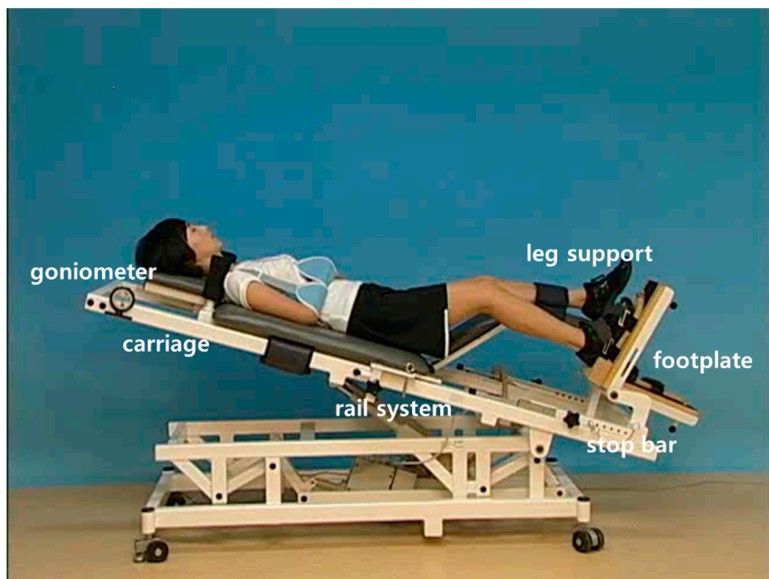

**Figure 1.** Sliding rehabilitation machine.

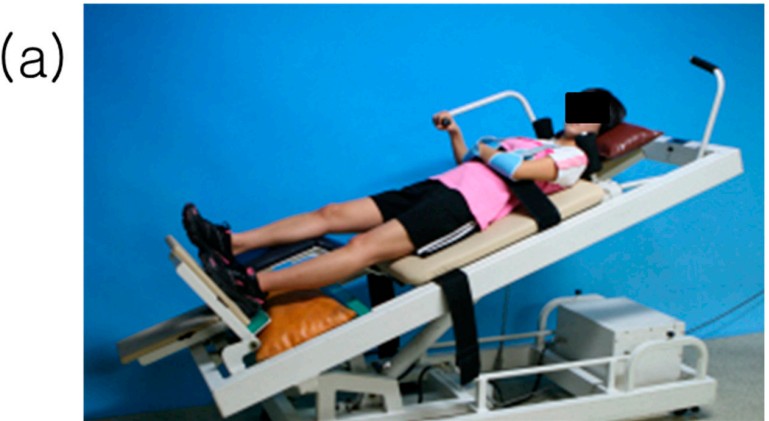

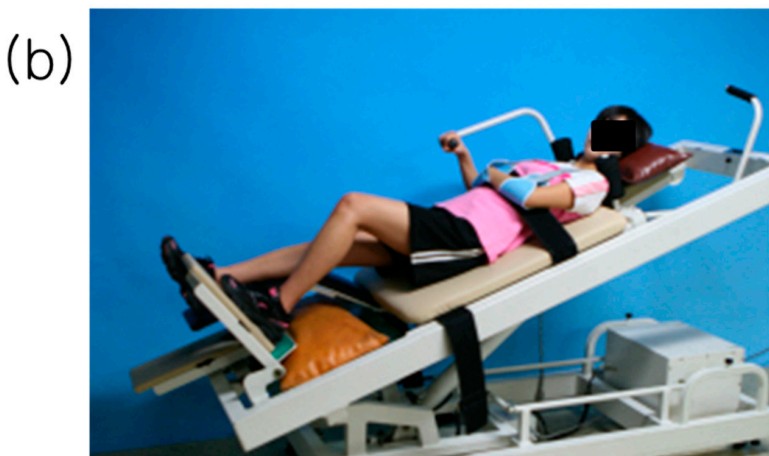

**Figure 2.** A patient is performing left lower extremity strengthening using the sliding rehabilitation machine: (**a**) left knee extension; (**b**) left knee flexion.

*2.2. Outcome Measure*

All patients underwent objective assessment before the training and at discharge. The SRM inclination angle was documented daily. The Berg Balance Scale (BBS) [20], Korean version of the Modified Barthel Index (K-MBI) [21,22], and manual muscle test (MMT) [23] were used as clinical tools to measure the level of impairment and function. All patients participated in SRM training on a daily basis, and if they were absent, their reasons were reviewed. Moreover, patients were assessed daily to ascertain if there were side effects or problems with the SRM training.

*2.3. Statistical Analysis*

Descriptive statistics were reported for all variables. For the outcome measure, paired *t*-tests were used to compare admission and discharge scores. The statistical software Statistical Package for the Social Sciences (SPSS Inc., Chicago, IL, USA), version 20.0, was used, and the *p*-value (alpha) of $< 0.05$ was considered significant.

Ethical approval was obtained from the institutional ethics committee.

## 3. Results

A total of 30 post-stroke patients with severe cognitive impairment who received additional SRM training during the admission period were included in the study. Of the patients, 12 were men and 18 were women, with 10 and 20 right and left hemisphere infarctions, respectively. The mean age was 74.3 (range 55 to 87) years, the mean time from stroke onset to admission to the rehabilitation department was 15.8 (range 5 to 29) days, and the mean duration of the training was 29.2 days. At baseline, the mean K-MBI was 8.0 (range 0 to 38), and the mean BBS was 5.1 (range 0 to 36). Patients' characteristics and the means of clinical tests before the training are shown in Tables 1 and 2. The baseline K-MMSE of the 30 patients was 1.6 points on average, indicating that patients participating in this study had severe cognitive impairment.

**Table 1.** General characteristics of subjects.

| Variables | |
|---|---|
| Patients (number) | 30 |
| Age (years) | $74.3 \pm 8.5$ |
| Sex (Male/Female) | 12:18 |
| Location(Rt./Lt.) | 10:20 |
| Training day | $29.2 \pm 7.9$ |
| Time from stroke to inclusion (days) | $15.8 \pm 6.0$ |
| K-MMSE | $1.6 \pm 2.6$ |
| NIHSS | $16.9 \pm 6.0$ |

Values are mean $\pm$ standard deviation. K-MMSE, Korean version of the Mini–Mental State Examination.

**Table 2.** Measurement values of clinical parameters.

| Clinical parameters | Baseline | Discharge | *p* |
|---|---|---|---|
| Angle of inclination | $6.7 \pm 4.6$ | $12.6 \pm 7.3$ * | 0.000 |
| BBS | $5.1 \pm 9.1$ | $15.4 \pm 17.4$ * | 0.006 |
| K-MBI | $8.0 \pm 9.2$ | $28.2 \pm 22.8$ * | 0.000 |
| MMT$_{HF}$ | $1.20 \pm 1.24$ | $1.70 \pm 1.32$ | 0.136 |
| MMT$_{KE}$ | $1.20 \pm 1.32$ | $1.67 \pm 1.54$ | 0.213 |
| MMT$_{ADF}$ | $1.13 \pm 1.30$ | $1.37 \pm 1.45$ | 0.515 |
| MMT$_{GTDF}$ | $1.13 \pm 1.30$ | $1.37 \pm 1.45$ | 0.515 |
| MMT$_{APF}$ | $1.17 \pm 1.29$ | $1.57 \pm 1.43$ | 0.260 |

Values are mean $\pm$ standard deviation; * $p < 0.05$ for change from admission to discharge; BBS, Berg Balance Scale; K-MBI, Korean version of the Modified Barthel Index; MMT, manual muscle test; HF, hip flexion; KE, knee extension; ADF, ankle dorsiflexion; GTDF, great toe dorsiflexion; APF, ankle plantar flexion.

### 3.1. Practical Aspects of the Use of SRM

For the 30 patients, 1736 sessions were performed out of 1754 scheduled sessions during the admission period. The performance rate was 98.9%. The causes of absence from the training were knee pain (5 times), sleep disturbance or depressive mood (7 times), dizziness (3 times), and schedule error (3 times) (Table 3).

**Table 3.** Reasons for absence from the training.

| Reasons | |
| --- | --- |
| Knee pain | 5 |
| Sleep disturbance or depression | 7 |
| Schedule error | 3 |
| Dizziness | 3 |
| Total | 18 |

Values are numbers.

A typical training session lasted for about 30 min. During the SRM training, some patients were given a manual guide or encouraged with a voice cue or physical touch. The time taken to prepare the training equipment was according to the severity of the patient's neurologic impairment. If the patient needed support to move onto the SRM, more time was required to prepare for the training, but it was not more than 10 min in any case.

### 3.2. Training Program and Clinical Course

The training data are presented in Table 2. The average SRM angle at the beginning of the training was 6.7° and was increased to 12.6° at the end of the training. At discharge, patients showed improvement in the SRM inclination angle, BBS, MMT, and K-MBI, and the changes in the SRM inclination angle, BBS, and K-MBI were significant (Table 2).

### 3.3. Adverse Events

No serious adverse events occurred. Minor and temporary side effects included knee pain (5 times), dizziness (3 times), and nausea (3 times).

## 4. Discussion

The main findings of this study show that the SRM enables intensive and repetitive lower extremity strengthening in post-stroke patients with cognitive impairment and that the SRM is feasible and safe as part of an inpatient rehabilitation program.

This study indicates that the functional improvement is the result of many factors such as spontaneous recovery, conventional treatment, and the SRM. Therefore, the SRM's effect on functional recovery cannot be known exactly. Although this study does not provide an exact conclusion about the effect of SRM training regarding recovery rate or final outcome as compared to other conventional programs, our findings suggest a positive effect on that aspect and may guide further research.

In conventional training, a tilt table, robot-assisted training [24], body-weight-supported treadmill training [25], therapist-assisted training, and braces are used to achieve upright standing and to compensate for lower extremity and trunk weakness. The degree of neurological deficit due to stroke is very diverse, and the application of conventional training and active participation are limited for post-stroke patients with cognitive impairment due to stroke cognitive decline [26]. Therefore, it is difficult to apply robot-assisted and task-performing training, which has become popular in recent years [27,28]. On the contrary, exercise using the SRM is a very simple movement that flexes and extends the lower extremities and allows the patient to instinctively learn the training method with a few repetitions in a short time. Moreover, SRM training can be easily applied to patients at a relatively low cost.

This study was the first to apply SRM training to post-stroke patients with severe cognitive impairment. The patient participation rate was 98.9%, and there were no serious side effects, only minor and transient side effects such as local knee pain or dizziness and nausea. However, whether dizziness and nausea are side effects from SRM training or may have been influenced by the overall condition of the patient is still unclear. We have observed changes and improvements over a wide functional area of post-stroke patients with severely cognitive impairment in this study (Table 2).

Patients who have had a stroke with severe cognitive dysfunction usually receive manual therapy with hand guidance, physical therapy using a tilt table or automatic bicycle, but voluntary muscle movements are limited using these therapies. With the SRM, patients can participate actively in the supervision and instructions from the physical therapist before and after the training. In addition, the SRM training system is a machine that physical therapist can use effectively for rehabilitation because it requires less time and effort to compare traditional gait training.

Various effects can be expected from the SRM training. First, if the patient is unable to stand independently, they can gradually strengthen their leg muscles by adjusting the SRM inclination, which reduces the load. By unloading a part of the body, the patient can perform a resistive closed kinetic exercise and improve leg muscle strength by repeating knee extension and flexion voluntarily. Previous studies have shown that weight bearing and stepping significantly improve locomotor capacity, which in turn supports the effect of the SRM [29,30]. Second, by reducing the influence of gravity and the patient's weight, a patient's range of motion can be easily controlled. This may lead to an increase in the patient's self-perceived performance and reduce anxiety and muscle guarding during the exercise through lower extremity flexion. Even post-stroke patients with cognitive decline can be actively involved. Finally, additional benefits of SRM use may be related to the psychosocial aspect of rehabilitation. The SRM allowed the patients to become active participants in their physical therapy. With an increase in the inclination angle and the number of repetitions during the training, the patients' lower extremity strength improved and this may have motivated them. It was a positive way to obtain feedback on the improvement of leg muscle strength in patients who could not stand independently prior to the training. SRM use appeared to improve patients' self-esteem and compliance with the therapy and to motivate them toward standing independently.

However, it is not necessary to continue with the training if the strength and function of the lower limbs have sufficiently improved to perform stand-up and squatting exercises independently.

According to the patients' reports and personal interactions with them, their attitude had generally changed positively to training with the SRM. Regaining independent gait function is often considered a primary goal of post-stroke rehabilitation, and it is possible to start training early with the SRM, which may serve as a motivating factor.

Some limitations hinder the generalizability of our results. First, this study was based on a small sample size and was conducted at one tertiary hospital without a control group. Second, post-stroke patients with severe cognitive impairment were included, but the severity of their neurological impairment was varied. Third, the retrospective research design was a limitation.

## 5. Conclusions

Muscle strengthening with the SRM in post-stroke patients with severe cognitive impairment is readily applicable and safe when used as part of an inpatient rehabilitation program. SRM training with conventional rehabilitation in post-stroke patients can increase the total amount of training to promote motor recovery and reduce the amount of work required by physical therapists.

**Author Contributions:** Conceptualization, Y.-S.L.; methodology, A.R.K. and Y.-S.L.; validation, A.R.K. and Y.-S.L.; formal analysis, A.R.K. and Y.-S.L.; investigation, A.R.K. and Y.-S.L.; resources, A.R.K. and Y.-S.L.; data curation, A.R.K. and Y.-S.L.; writing—original draft preparation, A.R.K. and Y.-S.L.; writing—review and editing, A.R.K. and Y.-S.L.; visualization, A.R.K. and Y.-S.L.; supervision, Y.-S.L.

**Funding:** This research received no external funding.

**Conflicts of Interest:** The authors declare no conflict of interest.

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
