# Peer review of "Application of Sliding Rehabilitation Machine in Patients with Severe Cognitive Dysfunction after Stroke"

_applsci, doi:10.3390/app9050927_

Round 1
Reviewer 1 Report
Sliding rehabilitation machine (SRM) allows closed kinetic chain exercises of the hip, knee, and ankle and offers repeated weight-bearing exercises. The authors investigated to explore the feasibility of SRM training when included in intensive rehabilitation program for post-stroke patients with severe cognitive dysfunction.
The concept of this study is novel and very interesting from the clinical perspective for the treatment of stroke patients. However, there are major problems related to the data presented that do not fully support the conclusions. Accordingly, this study is not complete, and the authors have to provide more detailed and insightful data interpretations.
1. The title of “Feasibility of Sliding Rehabilitation Machine on Post-Stroke Patients with Severe Cognitive” is not clear title. It is difficult to understand what author wants to tell. Please rethink the title.
2. The sections in the Materials and Methods and Results are not very informative. Authors should write in detail. Need more explanation.
3. There is no patient information. How do we know the post-stroke patients with severe cognitive dysfunction? Is there any data? For an example pre-training and post-training data?
Author Response
Response to Reviewer 1 Comments
We sincerely thank to the reviewers for their constructive and valuable comments. We have thoroughly addressed their specific concerns and have introduced the suggested changes that were very helpful in improving the manuscript. The main changes in the revised manuscript are red font colour. We used “Track Changes” function in Microsoft Word. Point-by-point responses to the reviewers’ comments are provided below.
1. The title of “Feasibility of Sliding Rehabilitation Machine on Post-Stroke Patients with Severe Cognitive” is not clear title. It is difficult to understand what author wants to tell. Please rethink the title.
Response 1: As a result of deliberation, we changed the title as follows. “Application of Sliding Rehabilitation Machine in Patients with Severe Cognitive Dysfunction after stroke”
2. The sections in the Materials and Methods and Results are not very informative. Authors should write in detail. Need more explanation.
Response 2: Material and Method parts have been modified with more detailed description. A description of the subject of the study was added, and the composition of the Sliding Rehabilitation Machine and how the patients trained using the SRM were described in detail.
Thus, the contents of the text have been modified and added as shown below, and figure 2 has been added.
Line 75-76 : Namely, patients were enrolled this study, who had severe cognitive impairment.
Line 78-83 : The SRM consists of a rail system, a carriage and footplate to support the patient. By moving the stop bar, which is able to manipulate angle of carriage, up and down then, physical therapist can change the patient’s angle of the knee flexion and extension. The leg support is connected to carriage that can support non hemiplegic lower extremity (Figure 1)
Line 83-85: The patient lies on parallel carriage to exercise, which is repetitive knee flexion and extension. This training can be performed that bilateral lower extremities or unilateral lower extremity ( Figure 2)
3. There is no patient information. How do we know the post-stroke patients with severe cognitive dysfunction? Is there any data? For an example pre-training and post-training data?
Response 3: We assessed the cognitive function of the patient through the Korean version of the Mini-Mental State Examination ( K-MMSE ), a cognitive assessment tool. The higher the K-MMSE score, the better the cognitive function. Based on references 18, 19, when K-MMSE was 9 or less, we judged that there was severe cognitive dysfunction and enrolled the patient. Thus, only patients with severe cognitive impairment were enrolled in the stroke patients.
Line 73-76 : (2) presence of severe cognitive impairment: Korean version of the Mini-Mental State Examination (K-MMSE) ≤9 [18,19]. Namely, patients were enrolled this study, who had severe cognitive impairment.

Reviewer 2 Report
Thank you for submitting this paper for review
In general: I would recommend revising for English (grammar, sentence structuring)
Specific
Line 13: the phrase “is readily applicable in stroke patient” needs grammar revision
Line 14: need to in line 12 put (SRM)
Line 17: what do you mean by repeated weight-bearing exercises? You note that closed kinetic chain exercises in the first half of the sentence. So is this redundancy?
Line 22: confused about the time frame. Did they exercise for 3-4 weeks from admission or something different?
Line 23: lowercase manual
Line 25: delete actually
Line 29-30: in patients who have experienced an ischemic stroke with severe cognitive dysfunction
Line 34: replace the with a
Line 35: try to use person first language. So change stroke patients to patients who have had a stroke
Line 35: delete that is and replace with ranging from mild to severe
Line 37: may reduce disability and may help to restore….
Line 38: recommend deleting the portion of the sentence after the 5].
Line 39-40: Suggest revising. A primary goal of patients after a stroke is to restore deficits in gait.
Line 40: Low or Lower?
Keep looking for places where you should change to first person language
Ex: stroke patients to patients who have had a stroke
Line 57: Should food and drug administration be capitalized? Is it a proper name?
Line 68: delete attack
Line 75: should avoid using “We”. This sentence is a bit of a run on too. Should be revised into a couple sentences.
Line 78: needs revision for grammar
Line 102: might want to say that alpha was set at 0.05
Line 106: …patients who were post-stroke with severe….
Line 125: table 2
I am not understanding what types of activities or exercises were performed during the training sessions
Would like to see more detail explaining what was done
Author Response
Response to Reviewer 2 Comments
We sincerely thank to the reviewers for their constructive and valuable comments. We have thoroughly addressed their specific concerns and have introduced the suggested changes that were very helpful in improving the manuscript. The main changes in the revised manuscript are red font colour. We used “Track Changes” function in Microsoft Word. Point-by-point responses to the reviewers’ comments are provided below.
In general: I would recommend revising for English (grammar, sentence structuring)
> We have reviewed and revised English grammar and sentence structures from beginning to end.
Specific
Line 13: the phrase “is readily applicable in stroke patient” needs grammar revision
> We modified as followed.
Line 12-14: This study illustrates that the use of Sliding Rehabilitation Machine (SRM) for early intensive muscle strengthening is applicable to stroke patient with severe cognitive dysfunction
Line 14: need to in line 12 put (SRM)
> The abbreviation was displayed and corrected.
Line 12-13 : Sliding Rehabilitation Machine (SRM)
Line 17: what do you mean by repeated weight-bearing exercises? You note that closed kinetic chain exercises in the first half of the sentence. So is this redundancy?
> The following changes have been made to avoid redundancy.
Line 16-17 : Sliding rehabilitation machine (SRM) allows closed kinetic chain exercises of the hip, knee, and ankle.
Line 22: confused about the time frame. Did they exercise for 3-4 weeks from admission or something different?
> The hospitalization period for rehabilitation was about 3-4 weeks, and training was conducted twice a day from Monday to Friday during the hospital stay.
We modified as followed.
Line 21-22 : Training with SRM was performed twice daily from Monday to Friday during they were admitted about 3 - 4 weeks.
Line 23: lowercase manual
> Modified as below according to lower case and upper case rule.
Line 23-24 : Berg Balance Scale (BBS), Manual Muscle Test (MMT) and Korean-Modified Barthel Index (K-MBI)
Line 25: delete actually
> Line 25 : ‘actually’ deleted.
Line 29-30: in patients who have experienced an ischemic stroke with severe cognitive dysfunction
We modified as followed.
Line 30 : in stroke patients with severe cognitive dysfunction.
Line 34: replace the with a
We modified as followed.
Line 34 : Stroke is a common cause of severe disability
Line 35: try to use person first language. So change stroke patients to patients who have had a stroke
We modified as followed.
Line 35 : Most patients who have had a stroke
Line 35: delete that is and replace with ranging from mild to severe
We modified as followed.
Line 35-36 : have varying degrees of neurological deficit ranging from mild to severe
Line 37: may reduce disability and may help to restore….
We modified as followed.
Line 37-38 : Proper rehabilitation may reduce disability and help to restore neurological impairment.
Line 38: recommend deleting the portion of the sentence after the 5].
> 5] The following sentence has been deleted.
Line 39-40: Suggest revising. A primary goal of patients after a stroke is to restore deficits in gait.
We modified it as you suggested.
Line 38-39 : A primary goal of patients after a stroke is to restore deficits in gait.
Line 40: Low or Lower?
Line 41: ‘Lower’ has been modified.
Keep looking for places where you should change to first person language
Ex: stroke patients to patients who have had a stroke
We modified as followed
Line 44 : In patients, who have had a stroke
Line 57: Should food and drug administration be capitalized? Is it a proper name?
We modified as followed
Line 58: Korea Food & Drug Administration (KFDA)
Line 68: delete attack >
Line 70 : We detelted attack
Line 75: should avoid using “We”. This sentence is a bit of a run on too. Should be revised into a couple sentences.
We modified as followed.
Line 78-83 : The SRM consists of a rail system, a carriage and footplate to support the patient. By moving the stop bar, which is able to manipulate angle of carriage, up and down then, physical therapist can change the patient’s angle of the knee flexion and extension. The leg support is connected to carriage that can support non hemiplegic lower extremity.
Line 78: needs revision for grammar
We modified as followed.
Line 82-83 : The leg support is connected to carriage that can support non hemiplegic lower extremity.
Line 102: might want to say that alpha was set at 0.05
We modified as followed.
Line 112 : the p value (alpha) of < 0.05 were considered significant.
Line 106: …patients who were post-stroke with severe….
We modified as followed.
Line 115 : A total of 30 patients who were post stroke with severe cognitive impairment
Line 125: table 2
Line 135 : table 2 notation has been corrected.
I am not understanding what types of activities or exercises were performed during the training sessions
Would like to see more detail explaining what was done
> The contents of the method have been added to the configuration of the SRM device, the role of each configuration, and the training method, and figure 2 has been added.
Line 78–85 : The SRM consists of a rail system, a carriage and footplate to support the patient. By moving the stop bar, which is able to manipulate angle of carriage, up and down then, physical therapist can change the patient’s angle of the knee flexion and extension. The leg support is connected to carriage that can support non hemiplegic lower extremity (Figure 1). The patient lies on parallel carriage to exercise, which is repetitive knee flexion and extension. This training can be performed that bilateral lower extremities or unilateral lower extremity (Figure 2).

Round 2
Reviewer 1 Report
The authors have responded appropriately to our original concerns. Only one thing that authors might need to put eye line in figure 1 same as Figure 2.
Author Response
Response to Reviewer 1 Comments_Round 2
Thank you for your careful reviews and comments.
The authors have responded appropriately to our original concerns. Only one thing that authors might need to put eye line in figure 1 same as Figure 2.
> Response : I put a shadow in the eye of Figure 1 as shown in Figure 2.
Thus, Figure 1 has been modified as shown below.

Reviewer 2 Report
Review Comments
Line 13: should use first person language. So: “patients who have had a stroke” instead of “stroke patient”
Line 21: grammar issue with the newly added language
Line 29: same as line 13
Look for changing to first person language throughout
Line 83: something is off grammar wise related to the word perform
Line 100: not sure that manual muscle test needs to be capitalized
Line 118: the new sentence needs revision. Not sure what is being said here.
Line 125: spell out minutes
Line 163: seems like a word or two is missing from the sentence (did not conclusive – seems like it is missing a sentence)
Line 181: I am confused here. Who is getting the manual therapy? The patients who have had a stroke? If so, why? If not, what are you trying to relate to here?
Author Response
Response to Reviewer 2 Comments_Round 2
Thank you for your careful reviews and comments.
Line 13: should use first person language. So: “patients who have had a stroke” instead of “stroke patient”
>We modified as followed.
Line 13-14 : patients who have had a stroke
Line 21: grammar issue with the newly added language
>We modified as followed.
Line 20-22 :Training with SRM was conducted twice a day from Monday to Friday during hospitalization for three to four weeks.
Line 29: same as line 13
>We modified as followed.
Line30: patients who have had a stroke
Look for changing to first person language throughout
>We modified as followed.
Line 46: patients who have had a stroke
Line 62: patients who have had a stroke
Line 83: something is off grammar wise related to the word perform
>We modified as followed.
Line 83-84: on which the patient is trained
Line 100: not sure that manual muscle test needs to be capitalized
> I checked it again and changed it to lowercase notation.
Line 101 : manual muscle test (MMT)
Line 118: the new sentence needs revision. Not sure what is being said here.
>We modified as followed.
Line 119-120 : The baseline K-MMSE of the 30 patients was 1.6 points on average, indicating that patients participating in this study had severe cognitive impairment.
Line 125: spell out minutes
>We modified as followed.
Line 126: minutes
Line 163: seems like a word or two is missing from the sentence (did not conclusive – seems like it is missing a sentence)
>We modified as followed.
Line 164: Although this study can not conclude the effect of training with SRM
Line 181: I am confused here. Who is getting the manual therapy? The patients who have had a stroke? If so, why? If not, what are you trying to relate to here?
> We would like to emphasize that patients who have had a stroke with severe cognitive impairment can not be actively involved in conventional rehabilitation program
We modified as followed.
Line 182-184 : Patients who have had a stroke with severe cognitive dysfunction usually receive manual therapy with hand guidance, physical therapy using a tilt table or automatic bicycle, but through these therapies, voluntary muscles movements are limited.
Thank you very much.
